# Robust Quantization: One Model to Rule Them All

**Moran Shkolnik** [†∘]  **Brian Chmiel** [†∘]  **Ron Banner** [†]

**Gil Shomron** [∘]  **Yury Nahshan** [†]  **Alex Bronstein** [∘]  **Uri Weiser** [∘]

[†]Habana Labs – An Intel company, Caesarea, Israel,
[∘]Department of Electrical Engineering - Technion, Haifa, Israel

{mshkolnik, bchmiel, rbanner, ynahshan}@habana.ai
gilsho@campus.technion.ac.il, bron@cs.technion.ac.il, uri.weiser@ee.technion.ac.il

## Abstract

Neural network quantization methods often involve simulating the quantization process during training, making the trained model highly dependent on the target bit-width and precise way quantization is performed. Robust quantization offers an alternative approach with improved tolerance to different classes of data-types and quantization policies. It opens up new exciting applications where the quantization process is not static and can vary to meet different circumstances and implementations. To address this issue, we propose a method that provides intrinsic robustness to the model against a broad range of quantization processes. Our method is motivated by theoretical arguments and enables us to store a single generic model capable of operating at various bit-widths and quantization policies. We validate our method's effectiveness on different ImageNet models. A reference implementation accompanies the paper.

## 1   Introduction

Low-precision arithmetic is one of the key techniques for reducing deep neural networks computational costs and fitting larger networks into smaller devices. This technique reduces memory, bandwidth, power consumption and also allows us to perform more operations per second, which leads to accelerated training and inference.

Naively quantizing a floating point (FP32) model to 4 bits (INT4), or lower, usually incurs a significant accuracy degradation. Studies have tried to mitigate this by offering different quantization methods. These methods differ in whether they require training or not. Methods that require training (known as *quantization aware training* or QAT) simulate the quantization arithmetic on the fly [Esser et al., 2019, Zhang et al., 2018, Zhou et al., 2016], while methods that avoid training (known as *post-training quantization* or PTQ) quantize the model after the training while minimizing the quantization noise [Banner et al., 2019, Choukroun et al., 2019, Finkelstein et al., 2019, Zhao et al., 2019].

But these methods are not without disadvantages. Both create models sensitive to the precise way quantization is done (e.g., target bit-width). Krishnamoorthi [2018] has observed that in order to avoid accuracy degradation at inference time, it is essential to ensure that all quantization-related artifacts are faithfully modeled at training time. Our experiments in this paper further assess this observation. For example, when quantizing ResNet-18 [He et al., 2015] with DoReFa [Zhou et al., 2016] to 4 bits, an error of less than 2% in the quantizer step size results in an accuracy drop of 58%.

There are many compelling practical applications where quantization-robust models are essential. For example, we can consider the task of running a neural network on a mobile device with limited resources. In this case, we have a delicate trade-off between accuracy and current battery life, which can be controlled through quantization (lower bit-width => lower memory requirements => less

energy). Depending on the battery and state of charge, a single model capable of operating at various quantization levels would be highly desirable. Unfortunately, current methods quantize the models to a single specific bit-width, experiencing dramatic degradations at all other operating points.

Recent estimates suggest that over 100 companies are now producing optimized inference chips [Reddi et al., 2019], each with its own rigid quantizer implementation. Different quantizer implementations can differ in many ways, including the rounding policy (e.g., round-to-nearest, stochastic rounding, etc), truncation policy, the quantization step size adjusted to accommodate the tensor range, etc. To allow rapid and easy deployment of DNNs on embedded low-precision accelerators, a single pre-trained generic model that can be deployed on a wide range of deep learning accelerators would be very appealing. Such a robust and generic model would allow DNN practitioners to provide a single off-the-shelf robust model suitable for every accelerator, regardless of the supported mix of data types, precise quantization process, and without the need to re-train the model on customer side.

In this paper, we suggest a generic method to produce robust quantization models. To that end, we introduce KURE — a KUrtosis REgularization term, which is added to the model loss function. By imposing specific kurtosis values, KURE is capable of manipulating the model tensor distributions to adopt superior quantization noise tolerance qualities. The resulting model shows strong robustness to variations in quantization parameters and, therefore, can be used in diverse settings and various operating modes (e.g., different bit-width).

This paper makes the following contributions: (i) we first prove that compared to the typical case of normally-distributed weights, uniformly distributed weight tensors have improved tolerance to quantization with a higher signal-to-noise ratio (SNR) and lower sensitivity to specific quantizer implementation; (ii) we introduce KURE — a method designed to uniformize the distribution of weights and improve their quantization robustness. We show that weight uniformization has no effect on convergence and does not hurt state-of-the-art accuracy before quantization is applied; (iii) We apply KURE to several ImageNet models and demonstrate that the generated models can be quantized robustly in both PTQ and QAT regimes.

## 2   Related work

**Robust Quantization.** Perhaps the work that is most related to ours is the one by Alizadeh et al. [2020]. In their work, they enhance the robustness of the network by penalizing the $L1-$norm of the gradients. Adding this type of penalty to the training objective requires computing gradients of the gradients, which requires running the backpropagation algorithm twice. On the other hand, our work promotes robustness by penalizing the fourth central moment (Kurtosis), which is differentiable and trainable through standard stochastic gradient methods. Therefore, our approach is more straightforward and introduces less overhead, while improving their reported results significantly (see Table 2 for comparison). Finally, our approach is more general. We demonstrate its robustness to a broader range of perturbations and conditions e.g., changes in quantization parameters as opposed to only changes to different bit-widths. In addition, our method applies to both post-training (PTQ) and quantization aware techniques (QAT) while [Alizadeh et al., 2020] focuses on PTQ.

**Quantization methods.** As a rule, these works can be classified into two types: post-training acceleration, and training acceleration. While post-training acceleration showed great successes in reducing the model weight's and activation to 8-bit, a more extreme compression usually involve with some accuracy degradation [Banner et al., 2019, Choukroun et al., 2019, Migacz, 2017, Gong et al., 2018, Zhao et al., 2019, Finkelstein et al., 2019, Lee et al., 2018, Nahshan et al., 2019]. Therefore, for 4-bit quantization researchers suggested fine-tuning the model by retraining the quantized model [Choi et al., 2018, Baskin et al., 2018, Esser et al., 2019, Zhang et al., 2018, Zhou et al., 2016, Yang et al., 2019, Gong et al., 2019, Elthakeb et al., 2019]. Both approaches suffer from one fundamental drawback - they are not robust to common variations in the quantization process or bit-widths other than the one they were trained for.

# 3 Model and problem formulation

Let $Q_\Delta(x)$ be a symmetric uniform $M$-bit quantizer with quantization step size $\Delta$ that maps a continuous value $x \in \mathbb{R}$ into a discrete representation

$$Q_\Delta(x) = \begin{cases} 2^{M-1}\Delta & x > 2^{M-1}\Delta \\ \Delta \cdot \left\lfloor \dfrac{x}{\Delta} \right\rceil & |x| \leq 2^{M-1}\Delta \\ -2^{M-1}\Delta & x < -2^{M-1}\Delta\,. \end{cases} \tag{1}$$

Given a random variable $X$ taken from a distribution $f$ and a quantizer $Q_\Delta(X)$, we consider the expected mean-squared-error (MSE) as a local distortion measure we would like to minimize, that is,

$$\text{MSE}(X, \Delta) = \mathbb{E}\left[(X - Q_\Delta(X))^2\right]\,. \tag{2}$$

Assuming an optimal quantization step $\tilde{\Delta}$ and optimal quantizer $Q_{\tilde{\Delta}}(X)$ for a given distribution $X$, we quantify the quantization sensitivity $\Gamma(X, \varepsilon)$ as the increase in $\text{MSE}(X, \Delta)$ following a small changes in the optimal quantization step size $\tilde{\Delta}(X)$. Specifically, for a given $\varepsilon > 0$ and a quantization step size $\Delta$ around $\tilde{\Delta}$ (i.e., $|\Delta - \tilde{\Delta}| = \varepsilon$) we measure the following difference:

$$\Gamma(X, \varepsilon) = \left| \text{MSE}(X, \Delta) - \text{MSE}(X, \tilde{\Delta}) \right|\,. \tag{3}$$

**Lemma 1** *Assuming a second order Taylor approximation, the quantization sensitivity $\Gamma(X, \varepsilon)$ satisfies the following equation (the proof in Supplementary Material 1.1):*

$$\Gamma(X, \varepsilon) = \left| \frac{\partial^2 MSE(X, \Delta = \tilde{\Delta})}{\partial^2 \Delta} \cdot \frac{\varepsilon^2}{2} \right|\,. \tag{4}$$

We use Lemma 1 to compare the quantization sensitivity of the Normal distribution with and Uniform distribution.

## 3.1 Robustness to varying quantization step size

In this section, we consider different tensor distributions and their robustness to quantization. Specifically, we show that for a tensor $X$ with a uniform distribution $Q(X)$ the variations in the region around $Q(X)$ are smaller compared with other typical distributions of weights.

**Lemma 2** *Let $X_U$ be a continuous random variable that is uniformly distributed in the interval $[-a, a]$. Assume that $Q_\Delta(X_U)$ is a uniform $M$-bit quantizer with a quantization step $\Delta$. Then, the expected MSE is given as follows (the proof in Supplementary Material 1.2):*

$$MSE(X_U, \Delta) = \frac{(a - 2^{M-1}\Delta)^3}{3a} + \frac{2^M \cdot \Delta^3}{24a}\,. \tag{5}$$

In Fig. 1(b) we depict the MSE as a function of $\Delta$ value for 4-bit uniform quantization. We show a good agreement between Equation 5 and the synthetic simulations measuring the MSE.

As defined in Eq. (3), we quantify the quantization sensitivity as the increase in MSE in the surrounding of the optimal quantization step $\tilde{\Delta}$. In Lemma 3 we will find $\tilde{\Delta}$ for a random variable that is uniformly distributed.

**Lemma 3** *Let $X_U$ be a continuous random variable that is uniformly distributed in the interval $[-a, a]$. Given an $M$-bit quantizer $Q_\Delta(X)$, the expected MSE is minimized by selecting the following quantization step size (the proof in Supplementary Material 1.3):*

$$\tilde{\Delta} = \frac{2a}{2^M + 1} \approx \frac{2a}{2^M}\,. \tag{6}$$

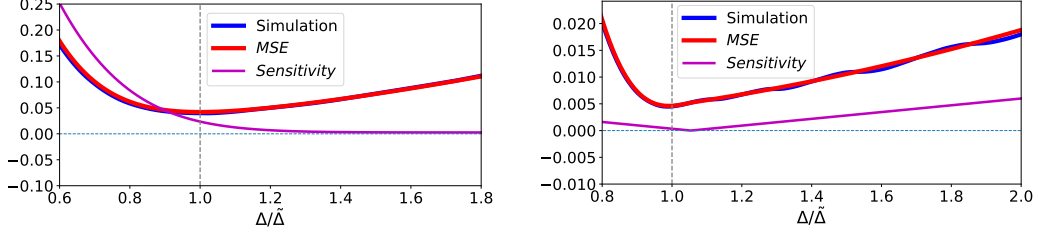

(a) **Optimal quantization of normally distributed tensors.** The first order gradient zeroes at a region with a relatively high-intensity 2nd order gradient, i.e., a region with a high quantization sensitivity $\Gamma(X_N, \varepsilon)$. This sensitivity zeros only asymptotically when $\Delta$ (and MSE) tends to infinity. This means that optimal quantization is highly sensitive to changes in the quantization process.

(b) **Optimal quantization of uniformly distributed tensors.** First and second-order gradients zero at a similar point, indicating that the optimum $\tilde{\Delta}$ is attained at a region where quantization sensitivity $\Gamma(X_U, \varepsilon)$ tends to zero. This means that optimal quantization is tolerant and can bear changes in the quantization process without significantly increasing the MSE.

Figure 1: Quantization needs to be modeled to take into account uncertainty about the precise way it is being done. The best quantization that minimizes the MSE is also the most robust one with uniformly distributed tensors (b), but not with normally distributed tensors (a). **(i) Simulation:** 10,000 values are generated from a uniform/normal distribution and quantized using different quantization step sizes $\Delta$. **(ii) MSE:** Analytical results, stated by Lemma 2 for the uniform case, and developed by [Banner et al., 2019] for the normal case - note these are in a good agreement with simulations. **(iii) Sensitivity:** second order derivative zeroes in the region with maximum robustness.

We can finally provide the main result of this paper, stating that the uniform distribution is more robust to modification in the quantization process compared with the typical distributions of weights and activations that tend to be normal.

**Theorem 4** *Let $X_U$ and $X_N$ be continuous random variables with a uniform and normal distributions. Then, for any given $\varepsilon > 0$, the quantization sensitivity $\Gamma(X, \varepsilon)$ satisfies the following inequality:*

$$\Gamma(X_U, \varepsilon) < \Gamma(X_N, \varepsilon), \tag{7}$$

*i.e., compared to the typical normal distribution, the uniform distribution is more robust to **changes** in the quantization step size $\Delta$ .*

**Proof:** In the following, we use Lemma 1 to calculate the quantization sensitivity of each distribution. We begin with the uniform case. We have presented in Lemma 2 the MSE$(X_U)$ as a function of $\Delta$. Hence, since we have shown in Lemma 3 that optimal step size for $X_U$ is $\tilde{\Delta} \approx \frac{a}{2^{M-1}}$ we get that

$$\Gamma(X_U, \varepsilon) = \left| \frac{\partial^2 \mathrm{mse}(X_U, \Delta = \tilde{\Delta})}{\partial^2 \Delta} \cdot \frac{\varepsilon^2}{2} \right| = \frac{2^{2M-1}(a - 2^{M-1}\tilde{\Delta}) + 2^{M-2}\tilde{\Delta}}{a} \cdot \frac{\varepsilon^2}{2} = \frac{\varepsilon^2}{4}. \tag{8}$$

We now turn to find the sensitivity of the normal distribution $\Gamma(X_N, \varepsilon)$. According to [Banner et al., 2019], the expected MSE for the quantization of a Gaussian random variable $N(\mu = 0, \sigma)$ is as follows:

$$\mathrm{MSE}(X_N, \Delta) \approx (\tau^2 + \sigma^2) \cdot \left[1 - \mathrm{erf}\left(\frac{\tau}{\sqrt{2}\sigma}\right)\right] + \frac{\tau^2}{3 \cdot 2^{2M}} - \frac{\sqrt{2}\tau \cdot \sigma \cdot e^{-\frac{\tau^2}{2 \cdot \sigma^2}}}{\sqrt{\pi}}, \tag{9}$$

where $\tau = 2^{M-1}\Delta$.

To obtain the quantization sensitivity, we first calculate the second derivative:

$$\frac{\partial^2 \mathrm{MSE}(X_N, \Delta = \tilde{\Delta})}{\partial^2 \Delta} = \frac{2}{3 \cdot 2^{2M}} - 2\,\mathrm{erf}\left(\frac{2^{M-1}\tilde{\Delta}}{\sqrt{2}\sigma}\right) - 2. \tag{10}$$

We have three terms: the first is positive but not larger than $\frac{1}{6}$ (for the case of $M = 1$); the second is negative in the range $[-2, 0]$; and the third is the constant $-2$. The sum of the three terms falls in the

range $[-4, -\frac{11}{6}]$. Hence, the quantization sensitivity for normal distribution is at least

$$\Gamma(X_N, \varepsilon) = \left| \frac{\partial^2 \text{MSE}(X_N, \Delta = \tilde{\Delta})}{\partial^2 \Delta} \cdot \frac{\varepsilon^2}{2} \right| \geq \frac{11\varepsilon^2}{12}. \tag{11}$$

This clearly establishes the theorem since we have that $\Gamma(X_N, \varepsilon) > \Gamma(X_U, \varepsilon)$ ∎

## 3.2 Robustness to varying bit-width sizes

Fig. 2 presents the minimum MSE distortions for different bit-width when normal and uniform distributions are optimally quantized. These optimal MSE values constitute the optimal solution of equations Eq. (5) and Eq. (9), respectively. Note that the optimal quantization of uniformly distributed tensors is superior in terms of MSE to normally distributed tensors at all bit-width representations.

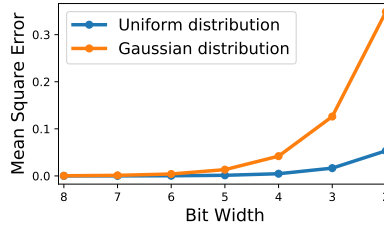

Figure 2: MSE as a function of bit-width for Uniform and Normal distributions. $\text{MSE}(X_U, \tilde{\Delta})$ is significantly smaller than $\text{MSE}(X_N, \tilde{\Delta})$.

## 3.3 When robustness and optimality meet

We have shown that for the uniform case optimal quantization step size is approximately $\tilde{\Delta} \approx \frac{2a}{2^M}$. The second order derivative is linear in $\Delta$ and zeroes at approximately the same location:

$$\Delta = \frac{2a}{2^M - \frac{1}{2^M}} \approx \frac{2a}{2^M}. \tag{12}$$

Therefore, for the uniform case, the optimal quantization step size in terms of $\text{MSE}(X, \tilde{\Delta})$ is generally the one that optimizes the sensitivity $\Gamma(X, \varepsilon)$, as illustrated by Fig. 1.

In this section, we proved that uniform distribution is more robust to quantization parameters than normal distribution. The robustness of the uniform distribution over Laplace distribution, for example, can be similarly justified. Next, we show how tensor distributions can be manipulated to form different distributions, and in particular to form the uniform distribution.

# 4 Kurtosis regularization (KURE)

DNN parameters usually follow Gaussian or Laplace distributions [Banner et al., 2019]. However, we would like to obtain the robust qualities that the uniform distribution introduces (Section 3). In this work, we use *kurtosis* — the fourth standardized moment — as a proxy to the probability distribution.

## 4.1 Kurtosis — The fourth standardized moment

The kurtosis of a random variable $\mathcal{X}$ is defined as follows:

$$\text{Kurt}[\mathcal{X}] = \mathbb{E}\left[ \left( \frac{\mathcal{X} - \mu}{\sigma} \right)^4 \right], \tag{13}$$

where $\mu$ and $\sigma$ are the mean and standard deviation of $\mathcal{X}$. The kurtosis provides a scale and shift-invariant measure that captures the shape of the probability distribution $\mathcal{X}$. If $\mathcal{X}$ is uniformly distributed, its kurtosis value will be 1.8, whereas if $\mathcal{X}$ is normally or Laplace distributed, its kurtosis values will be 3 and 6, respectively [DeCarlo, 1997]. We define "kurtosis target", $\mathcal{K}_T$, as the kurtosis value we want the tensor to adopt. In our case, the kurtosis target is 1.8 (uniform distribution).

## 4.2 Kurtosis loss

To control the model weights distributions, we introduce *kurtosis regularization* (KURE). KURE enables us to control the tensor distribution during training while maintaining the original model accuracy in full precision. KURE is applied to the model loss function, $\mathcal{L}$, as follows:

$$\mathcal{L} = \mathcal{L}_{\mathrm{p}} + \lambda \mathcal{L}_K , \tag{14}$$

$\mathcal{L}_p$ is the target loss function, $\mathcal{L}_K$ is the KURE term and $\lambda$ is the KURE coefficient. $\mathcal{L}_K$ is defined as

$$\mathcal{L}_K = \frac{1}{L} \sum_{i=1}^{L} \left| \mathrm{Kurt}\left[ \mathcal{W}_i \right] - \mathcal{K}_T \right|^2 , \tag{15}$$

where $L$ is the number of layers and $\mathcal{K}_T$ is the target for kurtosis regularization.

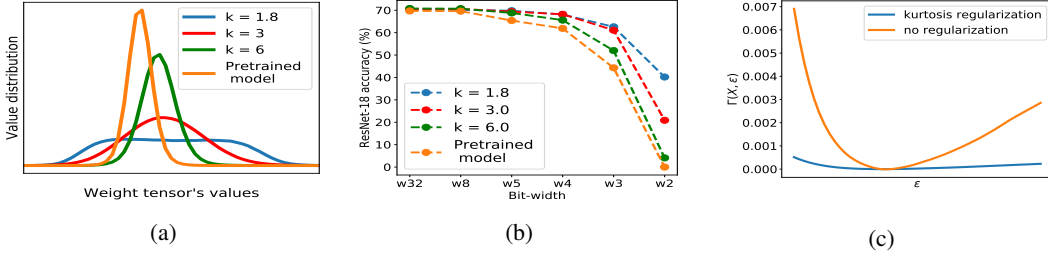

(a)                         (b)                         (c)

Figure 3: (a) Weights distribution of one layer in ResNet-18 with different $\mathcal{K}_T$. (b) Accuracy of ResNet-18 with PTQ and different $\mathcal{K}_T$. (c) Weights sensitivity $\Gamma(X, \varepsilon)$ in one ResNet-18 layer as a function of change in the step size from the optimal quantization step size ($\varepsilon = |\Delta - \tilde{\Delta}|$).

We train ResNet-18 with different $\mathcal{K}_T$ values. We observe improved robustness for changes in quantization step size and bit-width when applying kurtosis regularization. As expected, optimal robustness is obtained with $\mathcal{K}_T = 1.8$. Fig. 3c demonstrates robustness for quantization step size. Fig. 3b demonstrates robustness for bit-width and also visualizes the effect of using different $\mathcal{K}_T$ values. The ability of KURE to control weights distribution is shown in Fig. 3a.

## 5 Experiments

In this section, we evaluate the robustness KURE provides to quantized models. We focus on robustness to bit-width changes and perturbations in quantization step size. For the former set of experiments, we also compare against the results recently reported by Alizadeh et al. [2020] and show significantly improved accuracy. All experiments are conducted using Distiller [Zmora et al., 2019], using ImageNet dataset [Deng et al., 2009] on CNN architectures for image classification (ResNet-18/50 [He et al., 2015] and MobileNet-V2 [Sandler et al., 2018]).

### 5.1 Robustness towards variations in quantization step size

Variations in quantization step size are common when running on different hardware platforms. For example, some accelerators require the quantization step size $\Delta$ to be a power of 2 to allow arithmetic shifts (e.g., multiplication or division is done with shift operations only). In such cases, a network trained to operate at a step size that is not a power of two, might result in accuracy degradation. Benoit et al. [2017] provides an additional use case scenario with a quantization scheme that uses only a predefined set of quantization step sizes for weights and activations.

We measure the robustness to this type of variation by modifying the optimized quantization step size. We consider two types of methods, namely, PTQ and QAT. Fig. 4a and Fig. 4b show the robustness of KURE in ResNet50 for PTQ based methods. We use the LAPQ method [Nahshan et al., 2019] to find the optimal step size. In Fig. 4c and Fig. 4d we show the robustness of KURE for QAT based method. Here, we train one model using the DoReFa method [Zhou et al., 2016] combined with KURE and compare its robustness against a model trained using DoReFa alone. Both models are trained to the same target bit-width (e.g., 4-bit weights and activations). Note that a slight change of 2% in the quantization step results in a dramatic drop in accuracy (from 68.3% to less than 10%). In contrast, when combined with KURE, accuracy degradation turns to be modest

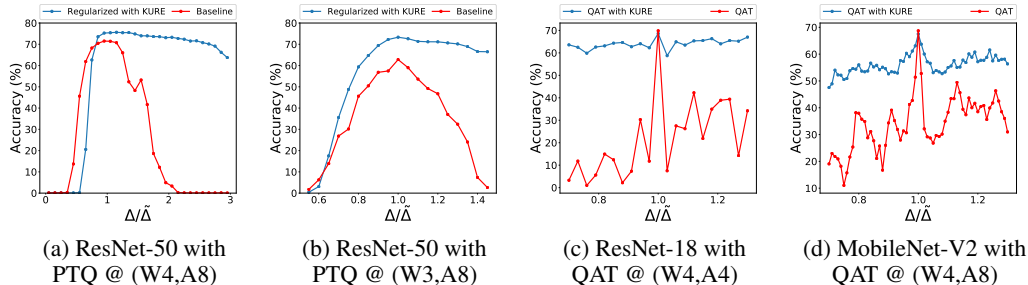

| (a) ResNet-50 with PTQ @ (W4,A8) | (b) ResNet-50 with PTQ @ (W3,A8) | (c) ResNet-18 with QAT @ (W4,A4) | (d) MobileNet-V2 with QAT @ (W4,A8) |

Figure 4: The network has been optimized (either by using LAPQ method as our PTQ method or by training using DoReFa as our QAT method) for step size $\tilde{\Delta}$. Still, the quantizer uses a slightly different step size $\Delta$. Small changes in optimal step size $\tilde{\Delta}$ of the weights tensors cause severe accuracy degradation in the quantized model. KURE significantly enhances the model robustness by promoting solutions that are more robust to uncertainties in the quantizer design. (a) and (b) show models quantized using PTQ method. (c) and (d) show models quantized with QAT method. @ (W,A) indicates the bit-width the model was quantized to.

## 5.2 Robustness towards variations in quantization bit-width

Here we test a different type of alteration. Now we focus on bit-width. We provide results related to QAT and PTQ as well as a comparison against [Alizadeh et al., 2020].

### 5.2.1 PTQ and QAT based methods

We begin with a PTQ based method (LAPQ - [Nahshan et al., 2019]) and test its performance when combined with KURE in Table 1. It is evident that applying KURE achieves better accuracy, especially in the lower bit-widths.

Table 1: KURE impact on model accuracy. (ResNet-18, ResNet-50 and MobileNet-V2 with ImageNet data-set)

| Model | Method | FP | 4 / FP | 3 / FP | 2 / FP | 6 / 6 | 5 / 5 | 4 / 4 | 3 / 3 |
|-------|--------|----|--------|--------|--------|-------|-------|-------|-------|
| | | | | | W/A configuration | | | | |
| ResNet-50 | No regularization | 76.1 | 71.8 | 62.9 | 10.3 | 74.8 | 72.9 | 70 | 38.4 |
| | **KURE regularization** | 76.3 | **75.6** | **73.6** | **64.2** | **76.2** | **75.8** | **74.3** | **66.5** |
| ResNet-18 | No regularization | 69.7 | 62.6 | 52.4 | 0.5 | 68.6 | 65.4 | 59.8 | 44.3 |
| | **KURE regularization** | 70.3 | **68.3** | **62.6** | **40.2** | **70** | **69.7** | **66.9** | **57.3** |
| MobileNet-V2 | No regularization | 71.8 | 60.4 | 31.8 | – | 69.7 | 64.6 | 48.1 | 3.7 |
| | **KURE regularization** | 71.3 | **67.6** | **56.6** | – | **70** | **66.9** | **59** | **24.4** |

Turning to QAT-based methods, Fig. 5 demonstrates the results with the LSQ quantization-aware method [Esser et al., 2019]. Additional results with different QAT methods can be found in the supplementary material.

### 5.2.2 A competitive comparison against [Alizadeh et al., 2020]

In Table 2 we compare our results to those reported by Alizadeh et al. [2020]. Our simulations indicate that KURE produces better accuracy results for all operating points (see Figure 2). It is worth mentioning that the method proposed by Alizadeh et al. [2020] is more compute-intensive than KURE since it requires second-order gradient computation (done through double-backpropagation), which has a significant computational overhead. For example, the authors mentioned in their work that their regularization increased time-per-epoch from 33:20 minutes to 4:45 hours for ResNet-18.

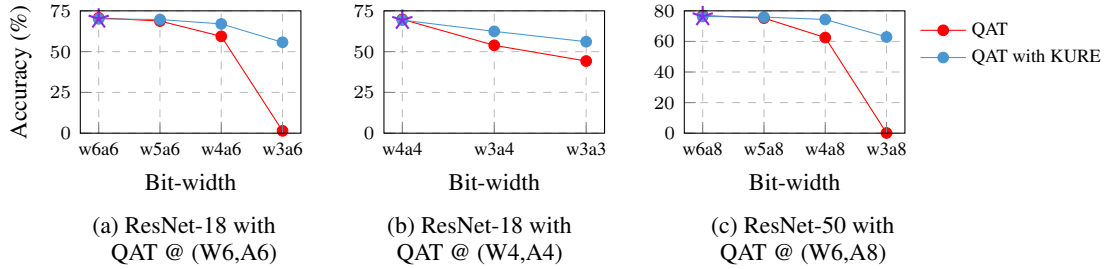

|  | (a) ResNet-18 with QAT @ (W6,A6) | (b) ResNet-18 with QAT @ (W4,A4) | (c) ResNet-50 with QAT @ (W6,A8) |

Figure 5: Bit-width robustness comparison of QAT model with and without KURE on different ImageNet architectures. We use LSQ method as our QAT method. The ⋆ is the original point to which the QAT model was trained. In (b) we change both activations and weights bit-width, while in (a) and (c) we change only the weights bit-width - which are more sensitive to quantization.

Table 2: Robustness comparison between KURE and [Alizadeh et al., 2020] for ResNet-18 on the ImageNet dataset.

|  |  | W/A configuration | | |
|---|---|---|---|---|
| Method | FP32 | 8 / 8 | 6 / 6 | 4 / 4 |
| L1 Regularization | 70.07 | 69.92 | 66.39 | 0.22 |
| L1 Regularization ($\lambda = 0.05$) | 64.02 | 63.76 | 61.19 | 55.32 |
| **KURE (Ours)** | **70.3** | **70.2** | **70** | **66.9** |

## 6  Summary

Robust quantization aims at maintaining a good performance under a variety of quantization scenarios. We identified two important use cases for improving quantization robustness — robustness to quantization across different bit-widths and robustness across different quantization policies. We then show that uniformly distributed tensors are much less sensitive to variations compared to normally distributed tensors, which are the typical distributions of weights and activations. By adding KURE to the training phase, we change the distribution of the weights to be uniform-like, improving their robustness. We empirically confirmed the effectiveness of our method on various models, methods, and robust testing scenarios.

This work focuses on weights but can also be used for activations. KURE can be extended to other domains such as recommendation systems and NLP models. The concept of manipulating the model distributions with kurtosis regularization may also be used when the target distribution is known.

## Broader Impact

Deep neural networks take up tremendous amounts of energy, leaving a large carbon footprint. Quantization can improve energy efficiency of neural networks on both commodity GPUs and specialized accelerators. Robust quantization takes another step and create one model that can be deployed across many different inference chips avoiding the need to re-train it before deployment (i.e., reducing $CO_2$ emissions associated with re-training).

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
