[Supplementary Material 1]

# Supplementary Material

## 1 Proofs from section: Model and problem formulation

### 1.1

**Lemma 1** *Assuming a second order Taylor approximation, the quantization sensitivity $\Gamma(X, \epsilon)$ satisfies the following equation:*

$$\Gamma(X, \epsilon) = \left| \frac{\partial^2 MSE(X, \Delta = \tilde{\Delta})}{\partial^2 \Delta} \cdot \frac{\epsilon^2}{2} \right| . \tag{1}$$

**Proof:** Let $\Delta'$ be a quantization step with similar size to $\tilde{\Delta}$ so that $|\Delta' - \tilde{\Delta}| = \epsilon$. Using a second order Taylor expansion, we approximate $\text{MSE}(X, \Delta')$ around $\tilde{\Delta}$ as follows:

$$\begin{aligned}
\text{MSE}(X, \Delta') = \text{MSE}(X, \tilde{\Delta}) &+ \frac{\partial \text{MSE}(X, \Delta = \tilde{\Delta})}{\partial \Delta}(\Delta' - \tilde{\Delta}) \\
&+ \frac{1}{2} \cdot \frac{\partial^2 \text{MSE}(X, \Delta = \tilde{\Delta})}{\partial^2 \Delta}(\Delta' - \tilde{\Delta})^2 + O(\Delta' - \tilde{\Delta})^3 .
\end{aligned} \tag{2}$$

Since $\tilde{\Delta}$ is the optimal quantization step for $\text{MSE}(X, \Delta)$, we have that $\frac{\partial \text{mse}(X, \Delta = \tilde{\Delta})}{\partial \Delta} = 0$. In addition, by ignoring order terms higher than two, we can re-write Equation (2) as follows:

$$\text{MSE}(X, \Delta') - \text{MSE}(X, \tilde{\Delta}) = \frac{1}{2} \cdot \frac{\partial^2 \text{MSE}(X, \Delta = \tilde{\Delta})}{\partial^2 \Delta}(\Delta' - \tilde{\Delta})^2 = \frac{\partial^2 \text{MSE}(X, \Delta = \tilde{\Delta})}{\partial^2 \Delta} \cdot \frac{\epsilon^2}{2} . \tag{3}$$

Equation (3) holds also with absolute values:

$$\Gamma(X, \epsilon) = \left| \text{MSE}(X, \Delta') - \text{MSE}(X, \tilde{\Delta}) \right| = \left| \frac{\partial^2 \text{MSE}(X, \Delta = \tilde{\Delta})}{\partial^2 \Delta} \cdot \frac{\epsilon^2}{2} \right| . \tag{4}$$

∎

### 1.2

**Lemma 2** *Let $X_U$ be a continuous random variable that is uniformly distributed in the interval $[-a, a]$. Assume that $Q_\Delta(X_U)$ is a uniform $M$-bit quantizer with a quantization step $\Delta$. Then, the expected MSE is given as follows:*

$$MSE(X_U, \Delta) = \frac{(a - 2^{M-1}\Delta)^3}{3a} + \frac{2^M \cdot \Delta^3}{24a} .$$

**Proof:** Given a finite quantization step size $\Delta$ and a finite range of quantization levels $2^M$, the quanitzer truncates input values larger than $2^{M-1}\Delta$ and smaller than $-2^{M-1}\Delta$. Hence, denoting by $\tau$ this threshold (i.e., $\tau \triangleq 2^{M-1}\Delta$), the quantizer can be modeled as follows:

$$Q_\Delta(x) = \begin{cases} \tau & x > \tau \\ \Delta \cdot \left\lfloor \dfrac{x}{\Delta} \right\rceil & |x| \leq \tau \\ -\tau & x < -\tau . \end{cases} \tag{5}$$

Therefore, by the law of total expectation, we know that

$$\begin{aligned}
\mathbb{E}\left[(x - Q_\Delta(x))^2\right] = \\
\mathbb{E}\left[(x - \tau)^2 \mid x > \tau\right] \cdot P\left[x > \tau\right] + \\
\mathbb{E}\left[\left(x - \Delta \cdot \lfloor \frac{x}{\Delta} \rceil\right)^2 \mid |x| \leq \tau\right] \cdot P\left[|x| \leq \tau\right] + \\
\mathbb{E}\left[(x + \tau)^2 \mid x < -\tau\right] \cdot P\left[x < -\tau\right] .
\end{aligned} \tag{6}$$

We now turn to evaluate the contribution of each term in Equation (6). We begin with the case of $x > \tau$, for which the probability density is uniform in the range $[\tau, a]$ and zero for $x > a$. Hence, the conditional expectation is given as follows:

$$\mathbb{E}\left[(x - \tau)^2 \mid x > \tau\right] = \int_\tau^a \frac{(x - \tau)^2}{a - \tau} \cdot dx = \frac{1}{3} \cdot (a - \tau)^2 . \tag{7}$$

In addition, since $x$ is uniformly distributed in the range $[-a, a]$, a random sampling from the interval $[\tau, a]$ happens with a probability

$$P\left[x > \tau\right] = \frac{a - \tau}{2a} . \tag{8}$$

Therefore, the first term in Equation (6) is stated as follows:

$$\mathbb{E}\left[(x - \tau)^2 \mid x > \tau\right] \cdot P\left[x > \tau\right] = \frac{(a - \tau)^3}{6a} . \tag{9}$$

Since $x$ is symmetrical around zero, the first and last terms in Equation (6) are equal and their sum can be evaluated by multiplying Equation (9) by two.

We are left with the middle part of Equation (6) that considers the case of $|x| < \tau$. Note that the qunatizer rounds input values to the nearest discrete value that is a multiple of the quantization step $\Delta$. Hence, the quantization error, $e = x - \Delta \cdot \lfloor \frac{x}{\Delta} \rceil$, is uniformly distributed and bounded in the range $[-\frac{\Delta}{2}, \frac{\Delta}{2}]$. Hence, we get that

$$\mathbb{E}\left[\left(x - \Delta \cdot \left\lfloor \frac{x}{\Delta} \right\rceil\right)^2 \mid |x| \leq \tau\right] = \int_{-\frac{\Delta}{2}}^{\frac{\Delta}{2}} \frac{1}{\Delta} \cdot e^2 de = \frac{\Delta^2}{12} . \tag{10}$$

Finally, we are left to estimate $P\left[|x| \leq \tau\right]$, which is exactly the probability of sampling a uniform random variable from a range of $2\tau$ out of a total range of $2a$:

$$P\left[|x| \leq \tau\right] = \frac{2\tau}{2a} = \frac{\tau}{a} . \tag{11}$$

By summing all terms of Equation (6) and substituting $\tau = 2^{M-1}\Delta$, we achieve the following expression for the expected MSE:

$$\mathbb{E}\left[(x - Q_\Delta(x))^2\right] = \frac{(a - \tau)^3}{3a} + \frac{\tau}{a}\frac{\Delta^2}{12} = \frac{(a - 2^{M-1}\Delta)^3}{3a} + \frac{2^M \Delta^3}{24a} . \tag{12}$$

∎

**1.3**

**Lemma 3** *Let $X_U$ be a continuous random variable that is uniformly distributed in the interval $[-a, a]$. Given an $M$-bit quantizer $Q_\Delta(X)$, the expected MSE $\mathbb{E}\left[(X - Q_\Delta(X))^2\right]$ is minimized by selecting the following quantization step size:*

$$\tilde{\Delta} = \frac{2a}{2^M \pm 1} \approx \frac{2a}{2^M} \,. \tag{13}$$

**Proof:** We calculate the roots of the first order derivative of Equation (12) with respect to $\Delta$ as follows:

$$\frac{\partial \text{MSE}(X_U, \Delta)}{\partial \Delta} = \frac{1}{a}\left(2^{M-3}\Delta^2 - 2^{M-1}\left(a - 2^{M-1}\Delta\right)^2\right) = 0 \,. \tag{14}$$

Solving Equation (14) yields the following solution:

$$\tilde{\Delta} = \frac{2a}{2^M \pm 1} \approx \frac{2a}{2^M} \,. \tag{15}$$

∎

**1.4  Hyper parameters to reproduce the results in Section 5- Experiments**

In the following section we describe the hyper parameters used in the experiments section. A fully reproducible code accompanies the paper.

**1.4.1  Hyper parameters for Section 5.1- Robustness towards variations in quantization step size**

In Table 1 we describe the hyper-parameters used in Fig. 4a and Fig. 4b in section 5.1 in the paper. We apply KURE on a pre-trained model from torch-vision repository and fine-tune it with the following hyper-parameters. When training phase ends we quantize the model using PTQ (Post Training Quantization) quantization method. All the other hyper-parameters like momentum and w-decay stay the same as in the pre-trained model.

Table 1: Hyper parameters for the experiments in section 5.1 - Robustness towards variations in quantization step size using PTQ methods

| arch | kurtosis target ($\mathcal{K}_T$) | KURE coefficient ($\lambda$) | initial lr | lr schedule | batch size | epochs | fp32 accuracy |
|---|---|---|---|---|---|---|---|
| ResNet-50 | 1.8 | 1.0 | 1e-3 | decays by a factor of 10 every 30 epochs | 128 | 50 | 76.4 |

In Table 2 we describe the hyper-parameters used in Fig. 4c and Fig. 4d in section 5.1 in the paper. We combine KURE with QAT method during the training phase with the following hyper-parameters.

**1.4.2  Hyper parameters for Section 5.2- Robustness towards variations in quantization bit-width**

In Table 3 we describe the hyper-parameters used in Table 1 in section 5.2.1 in the paper. We apply KURE on a pre-trained model from torch-vision repository and fine-tune it with the following hyper-parameters.

In Table 4 we describe the hyper-parameters used in Fig. 5 in section 5.2.1 in the paper. We combine KURE with QAT method during the training phase with the following hyper-parameters.

Table 2: Hyper parameters for experiments in section 5.1 - Robustness towards variations in quantization step size using QAT methods

| arch | QAT method | quantization settings (W/A) | kurtosis target ($\mathcal{K}_T$) | KURE co-efficient ($\lambda$) | initial lr | lr schedule | batch size | epochs | acc |
|---|---|---|---|---|---|---|---|---|---|
| ResNet-18 | DoReFa | 4 / 4 | 1.8 | 1.0 | 1e-4 | decays by a factor of 10 every 30 epochs | 256 | 80 | 68.3 |
| MobileNet-V2 | DoReFa | 4 / 8 | 1.8 | 1.0 | 5e-5 | lr decay rate of 0.98 per epoch | 128 | 10 | 66.9 |

Table 3: Hyper parameters for experiments in section 5.2 - Robustness towards variations in quantization bit-width using PTQ methods

| architecture | kurtosis target ($\mathcal{K}_T$) | KURE coefficient ($\lambda$) | initial lr | lr schedule | batch size | epochs | fp32 accuracy |
|---|---|---|---|---|---|---|---|
| ResNet-18 | | | | decays by a factor of 10 every 30 epochs | 256 | 83 | 70.3 |
| ResNet-50 | 1.8 | 1.0 | 0.001 | | 128 | 49 | 76.4 |
| MobileNet-V2 | | | | | 256 | 83 | 71.3 |

Table 4: Hyper parameters for experiments in section 5.2 - Robustness towards variations in quantization bit-width using QAT methods

| arch | QAT method | quantization settings (W/A) | kurtosis target ($\mathcal{K}_T$) | KURE co-efficient ($\lambda$) | initial lr | lr schedule | batch size | epochs | acc |
|---|---|---|---|---|---|---|---|---|---|
| ResNet-18 | LSQ | 6 / 6 | | | | decays by a factor of 10 every 20 epochs | 128 | 60 | 70.1 |
| ResNet-18 | LSQ | 4 / 4 | 1.8 | 1.0 | 1e-3 | | 128 | 60 | 69.3 |
| ResNet-50 | LSQ | 6 / 8 | | | | | 64 | 50 | 76.5 |

## 1.5 Robustness towards variations in quantization bit-width- additional results

In Fig. 5 in the paper we demonstrated robustness to variations in quantization bit-width of QAT models. we used LSQ method as our QAT model. In Fig. 1 we demonstrate the improved robustness with different QAT methods (DoReFa and LSQ) and ImageNet models.

(a) ResNet-18 with DoReFa @ (W6,A6)

(b) ResNet-18 with DoReFa @ (W5,A8)

(c) ResNet-50 with LSQ @ (W4,A8)

Figure 1: Bit-width robustness comparison of QAT model with and without KURE on different ImageNet architectures. The ⋆ is the original point to which the QAT model was trained.

## 1.6 Robustness towards variations in quantization step size- additional results

In section 5.1 in the paper, we explained the incentive to generate robust models for changes in the quantization step size. We mentioned that in many cases, accelerators support only a step size equal to a power of 2. In such cases, a model trained to operate at a step size different from a power of 2 value will suffer from a significant accuracy drop. Table 5 shows the accuracy results when the quantization step size is equal to a power of 2 compared to the optimal step size ($\tilde{\Delta}$) , for ImageNet models trained with and without KURE.

Table 5: KURE impact on model accuracy when rounding quantization step size to nearest power-of-2. (ResNet-18 and ResNet-50 with ImageNet data-set)

| Model | Method | W/A configuration | | | |
| | | 4 / FP | | 3 / FP | |
| | | $\Delta = \tilde{\Delta}$ | $\Delta = 2^N$ | $\Delta = \tilde{\Delta}$ | $\Delta = 2^N$ |
|---|---|---|---|---|---|
| ResNet-50 | No regularization | 71.8 | 63.6 | 62.9 | 53.2 |
| | **KURE regularization** | **75.6** | **74.2** | **73.6** | **71.6** |
| ResNet-18 | No regularization | 62.6 | 61.4 | 52.4 | 37.5 |
| | **KURE regularization** | **68.3** | **66.2** | **62.6** | **55.8** |

## 1.7 Statistical significance of results on ResNet-18/ImageNet trained with DoReFa and KURE

Table 6: Mean and standard deviation over multiple runs of ResNet-18 trained with DoReFa and KURE

| architecture | QAT method | quantization settings (W/A) | Runs | Accuracy, % (mean $\pm$ std) |
|---|---|---|---|---|
| ResNet-18 | DoReFa | 4 / 4 | 3 | $(68.4 \pm 0.09)$ |

Figure 2: The network has been trained for quantization step size $\tilde{\Delta}$. Still, the quantizer uses a slightly different step size $\Delta$. Small changes in optimal step size $\tilde{\Delta}$ cause severe accuracy degradation in the quantized model. KURE significantly enhances the model robustness by promoting solutions that are more robust to uncertainties in the quantizer design (ResNet-18 on ImageNet).



[Supplementary Material 2]

| | | | | | | | | | | | | | | |
|---|---|---|---|---|---|---|---|---|---|---|---|---|---|---|
| 0.025 | 0.001 | 0.003 | 0.003 | 0.000 | 0.001 | 0.000 | 0.001 | 0.001 | 0.000 | 0.002 | 0.000 | 0.001 | 0.000 | 0.000 |
| 0.001 | 0.030 | 0.005 | 0.006 | 0.005 | 0.004 | 0.000 | 0.002 | 0.001 | 0.000 | 0.001 | 0.002 | 0.002 | 0.001 | 0.001 |
| 0.003 | 0.005 | 0.018 | 0.000 | 0.004 | 0.002 | 0.001 | 0.000 | 0.002 | 0.001 | 0.001 | 0.001 | 0.002 | 0.000 | 0.002 |
| 0.003 | 0.006 | 0.000 | 0.020 | 0.005 | 0.001 | 0.002 | 0.004 | 0.001 | 0.001 | 0.000 | 0.001 | 0.001 | 0.001 | 0.003 |
| 0.000 | 0.005 | 0.004 | 0.005 | 0.030 | 0.005 | 0.005 | 0.004 | 0.002 | 0.002 | 0.001 | 0.002 | 0.000 | 0.000 | 0.003 |
| 0.001 | 0.004 | 0.002 | 0.001 | 0.005 | 0.023 | 0.001 | 0.003 | 0.003 | 0.002 | 0.002 | 0.002 | 0.001 | 0.000 | 0.008 |
| 0.000 | 0.000 | 0.001 | 0.002 | 0.005 | 0.001 | 0.014 | 0.005 | 0.003 | 0.002 | 0.000 | 0.002 | 0.001 | 0.001 | 0.003 |
| 0.001 | 0.002 | 0.000 | 0.004 | 0.004 | 0.003 | 0.005 | 0.016 | 0.005 | 0.002 | 0.000 | 0.001 | 0.001 | 0.000 | 0.002 |
| 0.001 | 0.001 | 0.002 | 0.001 | 0.002 | 0.003 | 0.003 | 0.005 | 0.027 | 0.003 | 0.000 | 0.001 | 0.002 | 0.001 | 0.001 |
| 0.000 | 0.000 | 0.001 | 0.001 | 0.002 | 0.002 | 0.002 | 0.002 | 0.003 | 0.019 | 0.000 | 0.002 | 0.002 | 0.000 | 0.002 |
| 0.002 | 0.001 | 0.001 | 0.000 | 0.001 | 0.002 | 0.000 | 0.000 | 0.000 | 0.000 | 0.029 | 0.006 | 0.005 | 0.000 | 0.001 |
| 0.000 | 0.002 | 0.001 | 0.001 | 0.002 | 0.002 | 0.002 | 0.001 | 0.001 | 0.002 | 0.006 | 0.029 | 0.005 | 0.000 | 0.002 |
| 0.001 | 0.002 | 0.002 | 0.001 | 0.000 | 0.001 | 0.001 | 0.001 | 0.002 | 0.002 | 0.005 | 0.005 | 0.049 | 0.001 | 0.002 |
| 0.000 | 0.001 | 0.000 | 0.001 | 0.000 | 0.000 | 0.001 | 0.000 | 0.001 | 0.000 | 0.000 | 0.000 | 0.001 | 0.024 | 0.012 |
| 0.000 | 0.001 | 0.002 | 0.003 | 0.003 | 0.008 | 0.003 | 0.002 | 0.001 | 0.002 | 0.001 | 0.002 | 0.002 | 0.012 | 0.099 |

[Supplementary Material 3 · qat_dorefa_sensitivity_resnet18_accu_for_different_scale_ratio.pdf]



Legend:
- ResNet18-DoReFa with KURE
- ResNet18-DoReFa

Y-axis: Accuracy (%)
X-axis: $\Delta/\Delta^*$