[Reviews · NeurIPS 2020]

Review 1

Summary and Contributions: This paper proposed a regularization method that can enhance the neural network accuracy when quantization is applied. The proposed method introduced Kurtosis loss, which is a simple term that motivates the weight distribution to be more uniform. The authors provided strong reasoning behind why a uniform shape is beneficial for the weight distribution when it is quantized. The authors also provided a performance comparison that highlighted the improved robustness against the quantization errors.

Strengths: - Clear explanation about why a uniform shape is beneficial for the weight - A simple method (= Kurtosis loss term) to enhance the robustness of the neural networks to the quantization errors.

Weaknesses: - There is little explanation about the impact of Kurtois to the activation quantization. - One of the most motivating use-cases of the proposed algorithm is the situation when the stepsize of the originally quantized neural net is not a power of two. In this situation, a solution that can easily modify the step size to become a power of two would be very desirable. Although the authors discussed this case in the paper, they did not provide enough supporting evidence on how much the proposed method can be helpful.

Correctness: I went through all the equations and derivations, and they look fine.

Clarity: Overall, this paper is well written and easy to follow.

Relation to Prior Work: This paper provides enough background.

Reproducibility: Yes

Additional Feedback: - Is there any particular reason of choosing Kurtosis over other statistical measure, such as coefficient of variation? [After reviewing the authors rebuttal, I would like to decrease my score for the following reasons] - The authors did not answer my question about the explanation about the impact of Kurtois to the activation quantization. Since KURE seems to be directly applied to the weights (but not to the activations, as shown in eq(15), it was not clear how KURE helps activation quantization to become robust to the change in quantization setting. Therefore, I asked this rather simple question with expectation of seeing some figures similar to Fig 3. But, very unfortunately, the authors provided some experimental results that are not highly related to this question. - For the 2nd question, the authors provided the additional experimental results, but they look confusing with limited explanation about their experimental settings. It seems that they repeated the PTQ experiments shown in sec 5.2.1. If this is the case, the new results does not look promising; For example, the new results show the accuracy without and with KURE for PTQ as 61.4% (wo KURE) vs 66.2% (w/ KURE) for ResNet18, while the reported results are 59.8% vs 66.9% (from Table 1). If the difference in the setting is that the power-of-two constraint is additionally applied to the new data, accuracy degradation due to it for the result with KURE (= 66.9% - 66.2% = 0.7%) is larger than the case without KURE (= 59.8% - 61.4% = -1.6%). On the other hand, ResNet50 somehow gets even better accuracy (=53.2% vs 71.6%) when the power-of-two constraints are applied compared to (38.4% vs 66.5%) -- here, again, one without KURE got higher accuracy "recovery". Therefore, it is not clear if KURE is indeed helpful for compensating the power-of-two constraint. If this shows the fact that KURE is not very effective to the power-of-two constraints (instead of the typical quantization), some of the implicit claims of the authors about the usecase ("... some accelerators require the quantization step size _x0001_ to be a power of 2 to allow arithmetic shifts") need to be modified. - For the 3rd question, the authors provided another option, entropy maximization, for data uniformization without clear justification on why they preferred KURE over it; the authors said it did not work -- why? Alos how much more complex it is? The problem here is that it is now unclear if KURE is the best choice. - Last but not least, I strongly think that the authors' answer to R3's question about the claim of the paper is not convincing. The authors provided an example usecase of "employing a 4-bit variant of the model when the battery is below 20% but the full precision one when the battery is over 80%", which is NOT the usual way of deploying models to the mobile devices. Since the memory footprint and the energy is highly restricted in the mobile devices, the quantized (e.g., 8-bit) models are deployed as the reduced-precision form. However, KURE seems to require full-precision weights as the mater copy to be deployed for dynamic quantization in other bits. Considering the overhead of keeping the full-precision weights in the memory and dynamically applying model quantization (different precision according to the battery life) is pretty high, this usecase does not seem to be very attractive.


Review 2

Summary and Contributions: This paper proposed 'robust quantization' by introducing a regularization term to make the model is not sensitive to step-wise of quantization. To motivate the proposed method, the authors provided a measurement of robustness to varying quantization step size and a theorem showing better robustness of uniform distribution compared with the normal distribution.

Strengths: 1. The theorem 4, saying that the uniform distribution is more robust to changes in the quantization step size makes sense. 2. The introduced regularization term, KURE, is able to make the distribution of weights like an uniform distribution according to Fig. 3, (a).

Weaknesses: In table 1, it can be observed that from 4-bit quantization to 3-bit quantization, the performance drops a lot. Can the authors provide any explanation about this?

Correctness: N/A

Clarity: The paper is easy to follow.

Relation to Prior Work: N/A

Reproducibility: Yes

Additional Feedback:


Review 3

Summary and Contributions: This article proposes a regularization method KURE that can make the weights uniformly distributed and improve the robustness of quantization. A sound theoretical analysis is provided, which proves that uniformly distributed weights are more robust than normally distributed weights. Authors did enough experiments on different data sets and different neural networks to support the effectiveness of the proposed method.

Strengths: The proposed Kure regularization method may be easy to applied to different models and architectures, which could be handy and useful.

Weaknesses: The experimental part is incomplete. No experimental parameter settings are provided, and no comprehensive comparison with the latest SOTA method is provided in the paper.

Correctness: I don't get the claim of the title of this paper "One model to rule them all" at all. No further expaination has been given on either "rule" or "all". Almost all quantization approach can be applied to different applications as long as enough data were provided given the context of DNN quantization; Second, the comparasion between KURE and the baseline model could be biased in Table 1. Since applying regularization, e.g. L2 regularization, is standard procedure in training DNN, it is unfair to compare with baseline approaches with no regularization in the experiments.

Clarity: Some part takes time to understand, the related work part had bad reading exprience.

Relation to Prior Work: Yes

Reproducibility: Yes

Additional Feedback: 1. 1. Line 114 proposes that the uniform distribution is more robust to the quantization process than the Gaussian distribution. However, formula (6) is derived based on the assumption that the X variable obeys a uniform distribution. The text does not mention what form the optimal quantization step size when the X variable obeys the Gaussian distribution is. So how this conclusion is reached lacks basis. 2. In line 105 of the text, only the picture b in Figure 1 is mentioned, which is the optimal quantization of the uniformly distributed tensor, but not the optimal quantization of the Gaussian distribution tensor in the a picture. The experimental results are incomplete. 3. The meaning of W_i in line 162 is not stated. 4. What is Kurt's formula in Eq 15? 5. What is the difference between L1 Regularization and L1 Regulation (\lambda = 0.05) in Table 2 I have checked the rebuttal, the author have cleared my concerns, so I raised my review score.


Review 4

Summary and Contributions: Quantization of weights is of great importance in deep learning as it leads to great savings in computational complexity. Many works simulate quantization as part of training and hope to have SGD compensate for quantization errors. This work proposes to regularize the weights so as to render them uniformly distributed. Indeed, uniformly distributed data suffers the least quantization noise.

Strengths: The authors have presented an excellent analysis where they theoretically show how uniformly distributed data suffers less quantization noise than normally distributed data. They introduce a sound metric, the kurtosis, which quantifies the uniformity of the tensor distribution. Regularizing this metric is a sound method to obtain desired distributions post-training.

Weaknesses: Two points I hope the authors can address. Does the kurtosis regularization impact the attainable loss in the original objective. Some analysis on that would significantly improve the contribution. Second, a lot of works are now looking at other number formats, such as low-precision floating-point and etc.. I do believe KURE can still be applied, but with an identification of another target kurtosis, since uniform data is no longer likely the most quantization friendly. Some analysis or discussion about that would also improve the quality of this paper.

Correctness: As far as I can tell, the paper is correct.

Clarity: The paper is clear and well written.

Relation to Prior Work: The overview of related works is well done.

Reproducibility: Yes

Additional Feedback: Comments post-rebuttal: Thanks to the authors for their responses. I am keeping my assessment unchanged.

[Author Response · NeurIPS 2020]

We thank all the reviewers for their helpful feedback, and for being unanimously positive about the submission: **R1:** *"The authors provided strong reasoning behind why a uniform shape is beneficial"*; **R2:** *"The paper is easy to follow"*; **R3:** *"Authors did enough experiments on different data sets and different neural networks"*; **R4:** *"The authors have presented an excellent analysis where they theoretically show how uniformly distributed data suffers less quantization noise"*. Below we address the main suggestions for improvements. Following Reviewer 1 suggestions, we ran two additional experiments that will be included in the final version along with accompanying code. If we address the reviewers' comments, we kindly ask that they adjust their scores to reflect their favorable opinion.

**R1:** *"There is little explanation about the impact of Kurtois to the activation quantization."* — Activations are typically less sensitive to quantization for vision workloads. Still, for the Neural Collaborative Filtering (NCF) model, we did some tests for activations with favorable results in all configurations: 63.07% vs. 62.09% (4 bits); 62.35% vs. 59.03% (3 bits); 56.06% vs. 36.46% (2 bits).

*"...a solution that can easily modify the step size to become a power of two would be very desirable."*— We conducted some new tests on ImageNet. When rounded to nearest power-of-two and in the case of 4-bit quantization, our method improves from 61.4% to 66.2%, and from 63.6% to 74.2% for ResNet18 and ResNet50, respectively. This becomes even more pronounced for 3-bit quantization going from 37.5% to 55.8%, and from 53.2% to 71.6% for ResNet18 and ResNet50, respectively.

*"Is there any particular reason of choosing Kurtosis over other statistical measure, such as coefficient of variation?"* — Kurtosis is a differentiable measure we can optimize to re-shape tensors into a uniform-like distribution. Entropy maximization is another option for data uniformization, which didn't work better than the Kurtisis measure but was more complicated to use in practice.

**R2:** *"In table 1, it can be observed that from 4-bit quantization to 3-bit quantization, the performance drops a lot. Can the authors provide any explanation about this?"* — Indeed, post-training quantization (PTQ) methods obtain mild degradation up to 4-bit quantization, which increases rapidly below that point. Our work pushes this boundary further offering a better trade-off for PTQ-based methods.

**R3** *"No experimental parameter settings are provided, and no comprehensive comparison with the latest SOTA method is provided in the paper."* — A detailed description of all experimental parameter settings is provided in the appendix (titled "hyperparameters to reproduce results") as well as a documented code. We compare against the results recently reported by Alizadeh et al. [2020] and demonstrate the improved robustness on two additional SOTA methods ([Nahshan et al., 2019] & [Esser et al., 2019]).

*"I don't get the claim of the title of this paper "One model to rule them all""* — We store a single set of weights ("one model") that can be applied with a large number of data-types ("to rule them all"). *"Almost all quantization approach can be applied to different applications as long as enough data were provided given the context of DNN quantization"* — Correct, but current methods do not show robustness when quantized to bit-widths other than the one they were trained for. In contrast, we allow for a single model to operate at various quantization levels (e.g., employ a 4-bit variant of the model when the battery is below 20% but the full precision one when the battery is over 80%.).

*"Second, the comparasion between KURE and the baseline model could be biased in Table 1. Since applying regularization, e.g. L2 regularization, is standard procedure in training DNN, it is unfair to compare with baseline approaches with no regularization in the experiments."* — "no regularization" means "no kurtosis regularization", but L2-regularization still applies. We will clarify that in the final version of the paper.

Additional comments: *"1. Line 114 proposes that the uniform distribution is more robust to the quantization process than the Gaussian distribution. However, formula (6) is derived based on ... a uniform distribution. "* — Equation 6 is only used to show that an optimal quantizer has a sensitivity of $\frac{\varepsilon^2}{4}$ for uniform inputs. Lines 124-132 prove that this sensitivity is larger than $\frac{\varepsilon^2}{4}$ for *any* quantizer with Gaussian inputs; *"2. Only the picture b in Figure 1 is mentioned"* — Thanks, we will include a reference to Fig. 1(a). *"3. The meaning of $W_i$ in line 162 is not stated."* — $W_i$ denotes the weight tensor of the i-th layer. *"4. What is Kurt's formula in Eq 15?"* — The Kurtosis formula is defined in Equation 13. *"5. What is the difference between L1 Regularization and L1 Regulation ($\lambda = 0.05$) in Table 2"* — In one configuration $\lambda$ is found through a grid-search, and in the other it is set to $\lambda = 0.05$.

**R4:** *"Does the kurtosis regularization impact the attainable loss in the original objective."* — FP accuracies before and after applying Kurtosis regularization are almost identical (See Table 1).

*"A lot of works are now looking at other number formats... I do believe KURE can still be applied, but with an identification of another target kurtosis."*— Thanks, we are currently working on other kurtosis targets to shape tensor distributions and make them more suitable for FP and binary quantizations.

[Meta-Review · NeurIPS 2020]

Two reviewers indicated accept and two reviewers indicated weak reject. The reviewers praised the good theoretical motivation and analysis, as well as the simple solution proposed in the paper, making it easy and widely applicable. Some concerns raised include missing experimental details, as well as the motivation of the method. While I agree with R1 that the rebuttal was not convincing, I believe that such method is still useful (eg when training a single model and deploying it on different hardware), and experimental results promising. Therefore, the paper is accepted.